# Extreme isotopic heterogeneity in Samoan clinopyroxenes constrains sediment recycling

Jenna V. Adams [1✉], Matthew G. Jackson[1,2], Frank J. Spera [1], Allison A. Price [1], Benjamin L. Byerly [1,3], Gareth Seward[1] & John M. Cottle[1]

Lavas erupted at hotspot volcanoes provide evidence of mantle heterogeneity. Samoan Island lavas with high $^{87}Sr/^{86}Sr$ (>0.706) typify a mantle source incorporating ancient subducted sediments. To further characterize this source, we target a single high $^{87}Sr/^{86}Sr$ lava from Savai'i Island, Samoa for detailed analyses of $^{87}Sr/^{86}Sr$ and $^{143}Nd/^{144}Nd$ isotopes and major and trace elements on individual magmatic clinopyroxenes. We show the clinopyroxenes exhibit a remarkable range of $^{87}Sr/^{86}Sr$—including the highest observed in an oceanic hot-spot lava—encompassing ~30% of the oceanic mantle's total variability. These new isotopic data, data from other Samoan lavas, and magma mixing calculations are consistent with clinopyroxene $^{87}Sr/^{86}Sr$ variability resulting from magma mixing between a high silica, high $^{87}Sr/^{86}Sr$ (up to 0.7316) magma, and a low silica, low $^{87}Sr/^{86}Sr$ magma. Results provide insight into the composition of magmas derived from a sediment-infiltrated mantle source and document the fate of sediment recycled into Earth's mantle.

[1] Department of Earth Science, University of California Santa Barbara, Santa Barbara, CA, USA. [2] Isotope Geochemistry Facility - Center for Mantle Zoology, University of California Santa Barbara, Department of Earth Science, Santa Barbara, CA, USA. [3] Thermo Fisher Scientific, Waltham, MA, USA. ✉email: adams@ucsb.edu

Subduction zones are the largest recycling systems on Earth where sediments, oceanic crust, mantle lithosphere, volatiles, and possibly blocks of continental crust return to the mantle. It is generally accepted that mantle plumes 'sample' such recycled materials leading to a diverse range of geochemical signatures reflected in ocean island basalts (OIBs)[1–3]. Mantle isotopic heterogeneities are inferred by examining global OIB lavas, but the scale, origin, and compositions of the sources and/or derivative melts remain under intense investigation. Many studies that have evaluated isotopic heterogeneity in OIBs have done so from the analysis of whole rock or bulk mineral separates[4–8], but these types of measurements obscure possible broader isotopic heterogeneity present within individual crystals or melt inclusions within a single lava. Previous work has shown $^{87}Sr/^{86}Sr$ and $^{143}Nd/^{144}Nd$ isotopic heterogeneity in lower oceanic crust clinopyroxene and plagioclase cumulate crystals, and another work has reported $^{87}Sr/^{86}Sr$ heterogeneity in olivine-hosted melt inclusions from a single lava sample—these are a couple of examples among many studies showing isotopic heterogeneity in individual crystals or melt inclusions[9–14]. Related to this study, recent work demonstrated intra- and inter-crystal $^{87}Sr/^{86}Sr$ heterogeneity in plagioclase in Samoan lavas with an enriched mantle 2 (EM2)[15] signature[16]. Collectively, these studies suggest that multiple isotopically distinct melts can contribute to a single lava, and thus magma mixing[13] is an essential part of igneous petrogenesis.

Subduction recycling of terrigenous sediments into the mantle has been cited to explain the EM2 geochemical signature recorded in lavas erupted at ocean island volcanoes[7,17]. Lavas with elevated $^{87}Sr/^{86}Sr$ (>0.706)[18] paired with low $^{143}Nd/^{144}Nd$, sampling an EM2 component, are relatively common among the volcanoes of the Samoan Islands in the South Pacific Ocean. Lavas erupted off the submarine flanks of Savai'i Island, western Samoa (Supplementary Fig. 1), have whole rock $^{87}Sr/^{86}Sr$ values up to 0.720469 (ref. [7]), plagioclase up to $0.7224 \pm 0.0003$ (2SE), and plagioclase intra-crystal variability of $^{87}Sr/^{86}Sr$ >5000 p.p.m.[16]. These studies have invoked two-endmember mantle mixing as the mechanism generating heterogeneity in Samoan lavas, suggesting up to 7% addition of a continental crust-like sediment in the Samoa plume source[16]. This study traces $^{87}Sr/^{86}Sr$ heterogeneity to a finer scale by obtaining coupled, high-precision Sr and Nd isotopes and major and trace elements on individual clinopyroxene crystals from the second-most (at the time of this study) enriched (high $^{87}Sr/^{86}Sr$) Samoan whole rock lava (whole rock $^{87}Sr/^{86}Sr =$ 0.718562).

Here we show that extreme $^{87}Sr/^{86}Sr$ heterogeneity observed in individual clinopyroxenes from a single geochemically extreme oceanic lava is consistent with magma mixing of at least two isotopically distinct melts—one basaltic and one trachytic. Further, the clinopyroxenes studied are found within one of the two Samoan lava samples in which Edwards et al.[16] measured plagioclase $^{87}Sr/^{86}Sr$ profiles from single porphyrocrysts. Consequently, a direct comparison of the extreme isotopic heterogeneity observed in the clinopyroxenes of this study and previously published plagioclase $^{87}Sr/^{86}Sr$ within a single hand specimen provides new constraints on the source of isotopic heterogeneity within the context of magma mixing and constraints from phase equilibria. Moreover, these new isotopic constraints help shed light on the origin of terrigenous sediment in the Samoan plume source and, thus, the deep-mantle residence of subducted terrigenous material in large low shear wave velocity provinces (LLSVPs)[19,20].

## Results and discussion

**An extreme EM2 sample from Samoa.** The studied sample is a 5 Ma submarine lava[21] ALIA-D115-18, dredged off the flanks of Savai'i Island, western Samoa (see Supplementary Fig. 1). This sample is a nepheline-normative trachyandesite with a bimodal crystal population. The larger crystal population is ≥25 μm in diameter and includes porphyrocrysts of clinopyroxene, plagioclase, and ilmenite as well as porphyroclasts consisting of the same three phases with modal abundances corresponding to a microgabbro. This study focuses on clinopyroxene porphyrocrysts, which were analyzed for major (electron microprobe) and trace elements (laser ablation inductively coupled mass spectrometry (LA-ICP-MS)) prior to dissolution and isotopic analysis by thermal ionization mass spectrometry (TIMS). Seventeen individual clinopyroxene crystals were analyzed for $^{87}Sr/^{86}Sr$ with resulting ratios ranging from 0.716833 (±0.000015, 2SE) to 0.723888 (±0.000015, 2SE), the highest value ever reported in an oceanic lava. Nine of the 17 crystals were also targeted for $^{143}Nd/^{144}Nd$ analysis, and single-crystal values range from 0.512238 (±0.000013, 2SE; $\varepsilon_{Nd} = -7.6$ calculated using $^{143}Nd/^{144}Nd_{chondrite} = 0.512630$ (ref. [22])) to 0.512357 (±0.000013, 2SE; $\varepsilon_{Nd} = -5.3$) (Supplementary Table 1). Together, the clinopyroxene Sr and Nd isotopes plot along an extension of the mantle array formed by global OIB, and including the new clinopyroxene data of this study, the $^{87}Sr/^{86}Sr$ variability in this single lava sample spans ~35% of the variability observed in all Samoan lavas, and ~30% of the entire oceanic mantle variability (Fig. 1).

**Generation of isotopic heterogeneity.** Data from clinopyroxene Sr and Nd isotopes from lava sample ALIA-D115-18 of this study, together with major and trace element and isotopic constraints from other Samoan whole rock lavas from the same submarine dredge as ALIA-D115-18 (referred to as the ALIA-D115 dredge), are used to constrain the composition of the magma mixing endmembers—including the extreme EM2 endmember—contributing to the isotopic variability observed in clinopyroxenes from ALIA-D115-18. This permits us to better define the origin and makeup of the high $^{87}Sr/^{86}Sr$ EM2 endmember in the Samoan plume, thereby constraining the petrogenesis of EM2 lavas.

In $^{87}Sr/^{86}Sr$ and $^{143}Nd/^{144}Nd$ space, addition of clinopyroxene data obtained in this study to the existing Samoan whole rock data requires a mixing line that exhibits more significant curvature (i.e. $r < 1$, where $r = [Nd/Sr]_M/[Nd/Sr]_S$[23]; where M [mafic] and S [silicic] represent the low- and high-$^{87}Sr/^{86}Sr$ endmembers, respectively[21]) than previous suggestions based on whole rock data alone (see ref. [4], Fig. 1). This implies that the Sr and Nd concentrations as well as the $^{87}Sr/^{86}Sr$ and $^{143}Nd/^{144}Nd$ values in the two endmembers need be re-evaluated. It should be noted, however, that the proposed mixing discussed here is not mixing between mantle sources, but mixing between magmas, and therefore differs from the approach in Jackson et al.[7]. Employing an inverse model approach to existing whole rock data from Samoa (see "Methods"), we argue for an EM2-derived mixing endmember that is not only high in $^{87}Sr/^{86}Sr$, but also trachytic in composition. Mixing of the EM2 trachytic magma with more primitive Samoan basalt-like compositions reproduces the mixing trends observed in ALIA-D115 whole rock lavas (Fig. 1) as well as the range of $^{87}Sr/^{86}Sr$ and $^{143}Nd/^{144}Nd$ in Samoa clinopyroxene.

The genesis of this high $^{87}Sr/^{86}Sr$, trachytic magma is an outstanding question and is further explored in the discussion and Supplementary Text. This aside, results of this study are consistent with the observation of Jackson et al.[7] who noted a distinct positive correlation between $^{87}Sr/^{86}Sr$ and $SiO_2$ concentration among Samoan whole rocks from the ALIA-D115 dredge haul, suggesting two-endmember mixing between a highly evolved (high silica, low MgO) endmember with high $^{87}Sr/^{86}Sr$

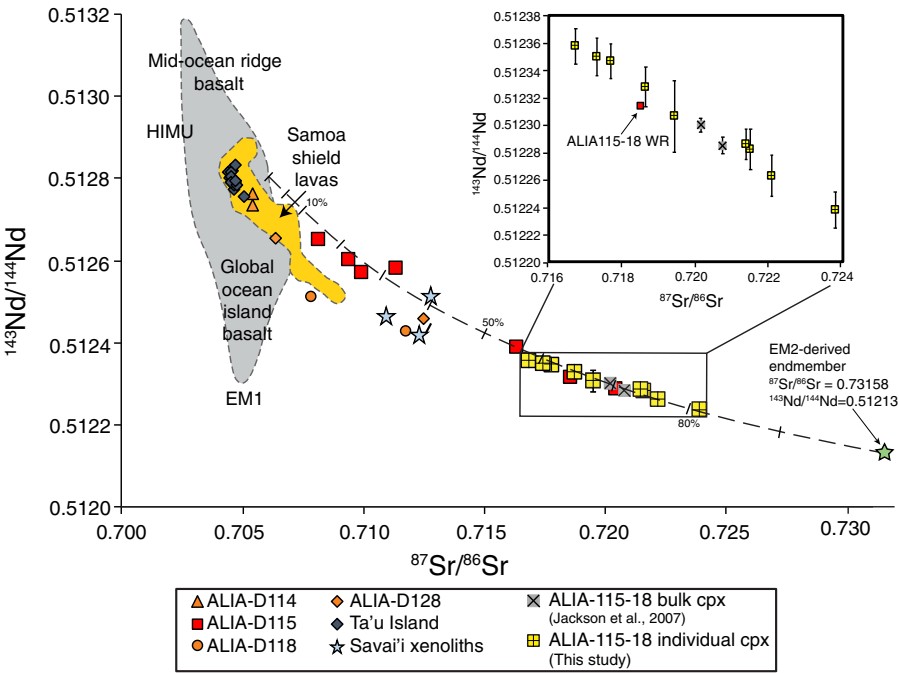

**Fig. 1** $^{143}Nd/^{144}Nd$ **versus** $^{87}Sr/^{86}Sr$ **of new ALIA-D115-18 single clinopyroxene data and prior data on the whole rock and bulk (100s of grains of) clinopyroxene.** Fields for Samoan shield lavas (excluding lavas from Savai'i Island), and global OIB and mid-ocean ridge basalts, are shown in orange and gray fields, respectively. Other Samoan shield lavas from the ALIA dataset (i.e., submarine shield lavas from Savai'i Island) are shown for reference as well (i.e. ALIA-D114: orange triangles, ALIA-D115: red squares, ALIA-D118: orange circles, and ALIA-D128 lavas: orange diamonds, and the new clinopyroxene data, denoted as yellow crossed squares, collected in this study)[7]. The inset shows a zoomed view of the clinopyroxene data in addition to the bulk clinopyroxene analyses (representing 100s of pooled grains; gray squares with an "x") and the ALIA-D115-18 whole rock (red square) published previously[7,8]. All error bars smaller than symbols except where shown and represent $2\sigma$ standard errors. Note that only nine of the 17 clinopyroxenes analyzed for $^{87}Sr/^{86}Sr$ also have $^{143}Nd/^{144}Nd$ data, so not all clinopyroxenes analyzed for $^{87}Sr/^{86}Sr$ can be shown on this figure. However, the nine clinopyroxenes analyzed for both Sr and Nd isotopes encompass the full range of $^{87}Sr/^{86}Sr$ observed in this study. Samoan shield lavas, ALIA lavas, and Savai'i xenolith compositions are from elsewhere[5–7,69]. The black dashed line is a theoretical magma mixing line derived in this study using binary mixing theory; mixing endmember compositions are provided in Supplementary Table 2, and methods for calculating endmembers are discussed in the text. Each hatch is a 10% increment except for the first two hatches, which are in 5% increments. The green star denotes the EM2-derived, high $^{87}Sr/^{86}Sr$, trachytic mixing endmember calculated in this study (see Supplementary Table 2).

and a less evolved (basaltic) endmember with low $^{87}Sr/^{86}Sr$. In addition, Edwards et al.[16] noted a negative correlation between $^{87}Sr/^{86}Sr$ and Sr concentration in plagioclase crystals suggesting the high $^{87}Sr/^{86}Sr$ (the "EM2" component) endmember likely experienced significant loss of Sr due to plagioclase fractionation prior to mixing, consistent with the "EM2" component being highly evolved prior to mixing. If extreme plagioclase fractionation of the magma with radiogenic $^{87}Sr/^{86}Sr$ did occur prior to mixing, then a negative correlation should also be observed in the clinopyroxene since they are a direct result of the mixing process. However, there is no clear correlation between clinopyroxene $^{87}Sr/^{86}Sr$ and Sr concentration, which can be best explained if plagioclase fractionation occurred after the onset of mixing. This conceptual model is consistent with MELTS[24–26] and Magma Chamber Simulator (MCS)[27] mixing modeling, which shows that clinopyroxene crystallization begins ~50 °C prior to plagioclase saturation upon mixing of a mafic, low $^{87}Sr/^{86}Sr$ melt and a silicic, high $^{87}Sr/^{86}Sr$ melt. This scenario is also consistent with the observation that the range of $^{87}Sr/^{86}Sr$ in the clinopyroxenes (0.71683–0.72389) exceeds that of the range of plagioclase $^{87}Sr/^{86}Sr$ values found in the same hand specimen (0.71791–0.72239) by Edwards et al.[16] (Fig. 2), suggesting that upon plagioclase crystallization, the system may have been more homogenized relative to an earlier time when clinopyroxene crystallization initiated.

Figure 3 shows geochemical data from the ALIA-D115 dredge whole rock lavas (red symbols) in addition to shield lavas (orange symbols) and pillow glasses (purple symbols) from neighboring Samoan islands and seamounts. Mixing is clearly defined by the ALIA-D115 dredge lavas in $^{87}Sr/^{86}Sr$ versus element concentration space (Fig. 3), and in major and select trace element space (Supplementary Figs. 2 and 3), spanning compositions varying from trachybasaltic to trachyandesitic in a single dredge: curved lines define mixing in the ALIA-D115 lavas, while the rest of the Samoan lavas show zero-slope fractional crystallization trends.

The Sr, Nd, and major element oxide concentrations, as well as the $^{87}Sr/^{86}Sr$ and $^{143}Nd/^{144}Nd$ of the two endmembers that anchor the ALIA-D115 mixing array are reconstructed using a mixing analysis. Note that we do not use the clinopyroxenes for reconstructing the endmembers because they show ill-defined mixing trends with respect to major and trace elements. This is likely due to (1) fractionation of trace elements during crystal growth and (2) the clinopyroxenes being zoned such that averaged spot analyses of trace element concentrations (determined by averaging multiple LA-ICP-MS spots on each crystal; see "Methods") will not correspond directly with grain surface major elements (obtained by extracting average major element concentrations from EPMA maps; see "Methods") and will thus produce spurious trends, especially when compared to radiogenic isotopes (which were measured on dissolved crystals). However, these issues will not affect the isotopes, which is why the mixing

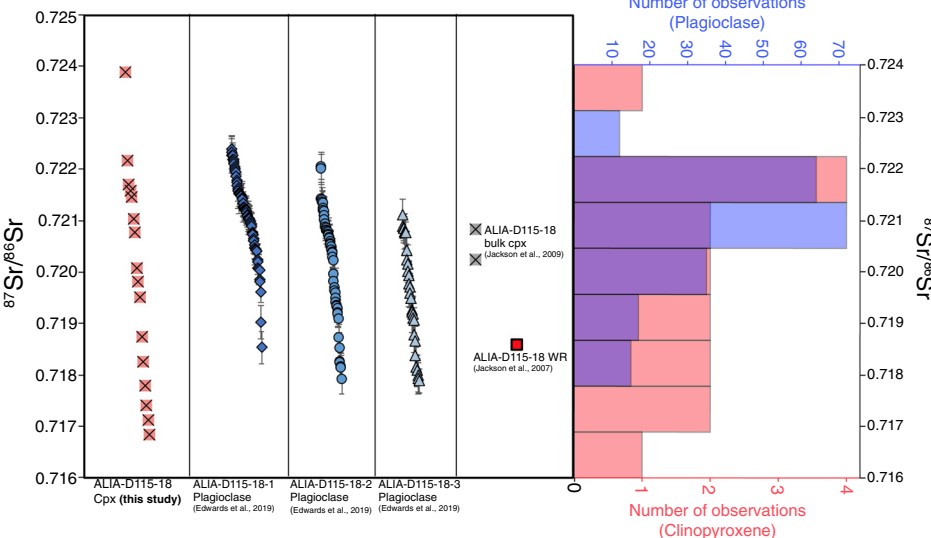

**Fig. 2** **$^{87}$Sr/$^{86}$Sr values for single clinopyroxene grains measured in this study with LA-ICP-MS spot analyses from three plagioclase grains from the same sample (ALIA-D115-18)[16].** Previous measurements of bulk clinopyroxene[8] (representing 100s of pooled grains; gray squares with an "x") from this same sample are shown together with the whole rock value[7] (where WR stand for whole rock; red square), clinopyroxenes measured in this study (pink squares with an "x"), and plagioclase analyses from Edwards et al.[16] (blue diamonds, circles, and triangles); all error bars smaller than symbols except where shown and represent 2σ standard errors. In the histogram, the number of clinopyroxene observations ($n = 17$; red bars and red figure axis label) span a larger range of $^{87}$Sr/$^{86}$Sr than the plagioclase $^{87}$Sr/$^{86}$Sr analyses[16] ($n = 157$; blue bars and blue figure axis label; error bars are 2 standard errors); where the clinopyroxene and plagioclase histograms overlap, a purple shade is used. The wider range of clinopyroxene $^{87}$Sr/$^{86}$Sr values compared with the plagioclase from the same sample is consistent with magma mixing modeling using MELTS[24-26] and the Magma Chamber Simulator[27]. The magma mixing models show that magma mixing between a mafic endmember and a more silicic, trachytic endmember (as derived in this study) precipitates clinopyroxene about 50 °C prior to plagioclase saturation and thus the mixing magmas may be slightly more homogenized at this point, explaining the reduced range of $^{87}$Sr/$^{86}$Sr exhibited by plagioclase (see Main text and "Methods" for more detail).

trend is evident for $^{87}$Sr/$^{86}$Sr and $^{143}$Nd/$^{144}$Nd. The method for defining the endmember compositions using the ALIA-D115 lava trends (i.e., to reconstruct the composition of both endmembers), in terms of major elements and Sr and Nd concentrations, requires initial estimates for the MgO content of the endmembers and linear regression analysis of the ALIA whole rock bulk compositions in $C_i$/MgO versus 1/MgO space (where $C_i$ is the concentration of a major element oxide or a trace element; see detailed description in the "Methods" section). MgO is estimated, ab initio, because it can be best constrained in both mixing endmembers given the limited variability at both ends of the mixing array formed by the ALIA-D115 lavas as observed when comparing SiO$_2$ versus MgO, for example (see Supplementary Fig. 2). In comparing SiO$_2$ versus MgO concentrations, the ALIA-D115 lavas show a linear mixing array that extends from a group of mafic Samoan shield lavas to a hypothetical endmember extending toward 0 wt% MgO. An averaged Samoan shield lava value is used for the MgO concentration of the mafic mixing endmember and we test three different MgO concentrations for the silicic mixing endmember from the lowest value observed in ALIA-D115 lavas (3.8 wt% MgO) to 0.5 wt% MgO (see "Methods" for more details). Further, according to mixing theory, $^{87}$Sr/$^{86}$Sr versus Nd/Sr concentration and $^{143}$Nd/$^{144}$Nd versus Sr/Nd concentration (companion plots to the hyperbolic mixing relation in $^{87}$Sr/$^{86}$Sr versus $^{143}$Nd/$^{144}$Nd space[23]) should give rise to linear arrays in a system described by two-component mixing. Thus, linear regressions were performed using these linear arrays, and given the derived Sr and Nd concentrations in each endmember, estimates of $^{87}$Sr/$^{86}$Sr and $^{143}$Nd/$^{144}$Nd were obtained for both endmembers (see "Methods" for more detail, Supplementary Figs. 4 and 5 for regressions, and Supplementary Table 2 for calculated endmember compositions). In addition, as proof of concept, this linear regression method was applied to two

other select trace elements (Nb, Th) pertinent to generating a ratio useful for tracking source mantle characteristics (Nb/Th[28]; see Supplementary Fig. 8 and Supplementary Table 2 for concentrations in the endmembers).

Given an initial assumption of 10 wt% and 0.5 wt% MgO in the low $^{87}$Sr/$^{86}$Sr (mafic) and high $^{87}$Sr/$^{86}$Sr (silicic) endmembers, respectively (see "Methods" for justification), we calculate 380 p.p.m. Sr and 63 p.p.m. Nd in the high $^{87}$Sr/$^{86}$Sr EM2 endmember magma (see Supplementary Table 2 for composition of two other high $^{87}$Sr/$^{86}$Sr endmember calculations with different starting MgO concentrations) and retrieve values of $^{87}$Sr/$^{86}$Sr = 0.73158 and $^{143}$Nd/$^{144}$Nd = 0.51213 ($\varepsilon_{Nd} = -9.8$); the calculated major element composition of this EM2 endmember magma, which is trachytic, is provided in Supplementary Table 2. We calculate 712 p.p.m. Sr and 48 p.p.m. Nd in the low $^{87}$Sr/$^{86}$Sr endmember and retrieve values of $^{87}$Sr/$^{86}$Sr = 0.70610 and $^{143}$Nd/$^{144}$Nd = 0.51280 (see Supplementary Table 2). Figures 1 and 3 illustrate mixing curves (black lines) using these values showing good agreement with the ALIA-D115 lavas and the clinopyroxenes of this study.

The percentage of mixing observed in $^{87}$Sr/$^{86}$Sr versus $^{143}$Nd/$^{144}$Nd space (Fig. 1), consistent with the other major element and isotope plots (Fig. 3 and Supplementary Figs. 2 and 3), suggests that the most silica-rich ALIA-D115 lava (sample ALIA-D115-21) is composed of ~70–80% of the calculated EM2 endmember melt and ~20–30% of the mafic melt endmember.

**Preservation of isotopic heterogeneity in the magma chamber.** Recent work has shown zoning in pyroxene to be a reliable indicator of magma mixing events[29,30]. Here, Mg and Fe zoning in clinopyroxene is used to assess magmatic residence times post magma mixing (where magmatic residence time corresponds to the time needed to eradicate anomalous compositional lamella in

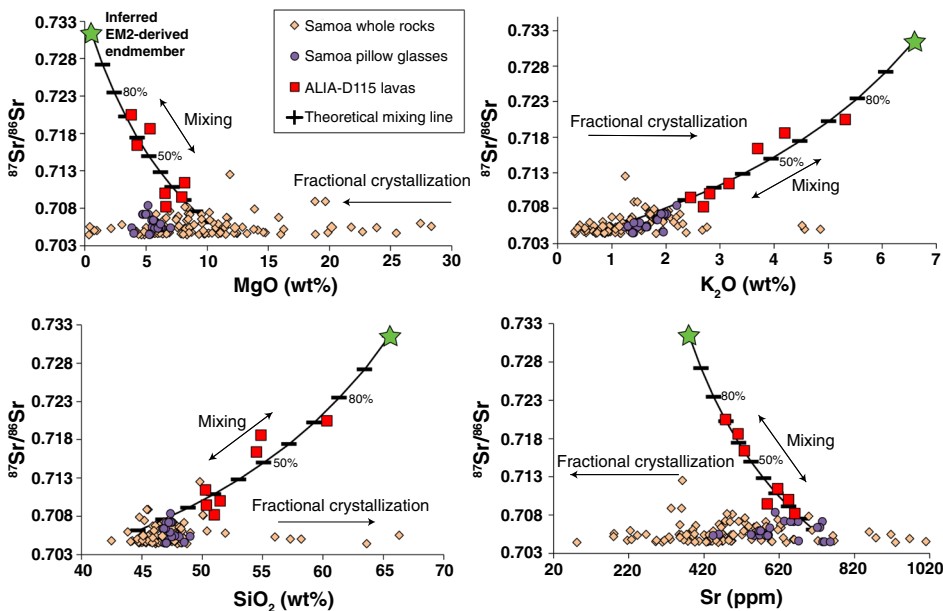

**Fig. 3 $^{87}$Sr/$^{86}$Sr versus element concentration showing theoretical mixing curves of inferred endmember compositions.** The direction of fractional crystallization is denoted by arrows. The green star denotes the EM2-derived mixing endmember composition (SiO$_2$ = 65.5 wt%, MgO = 0.5 wt%, Al$_2$O$_3$ = 17.1 wt%, FeO = 3.8 wt%, CaO = 2.7 wt%, TiO$_2$ = 1 wt%, and Na$_2$O + K$_2$O = 11 wt%; see Supplementary Table 2), which is calculated in this study (see "Methods"); the black hatched line shows the path of the calculated mixing curves, where each hatch increases by 10% mixing from the geochemically depleted mafic endmember to the more evolved EM2-derived endmember (black hatched line is derived from the same methodology as described briefly in Fig. 2 and in the "Methods"). Mixing is clearly defined by a curve with a non-zero slope, and is unlike the lavas that have been affected only by fractional crystallization (which have zero-slope trends). The ALIA-D115 lavas are represented by red squares, the other Samoan whole rock lavas by orange diamonds, and Samoan pillow glasses by purple circles. See Fig. 1 caption for data references; Samoa pillow glass data can be found in Workman et al.[6]. All errors bars smaller than symbols.

clinopyroxene), given relevant temperatures and diffusivities. The eradication of compositional anomalies in the form of thin growth lamella can be computed following a standard one-dimensional diffusion analysis[31] (details are presented in the "Methods" section). Attention is focused on the most mobile elements in clinopyroxene that exhibit zonation and for which diffusivity data exist at the inferred temperature range relevant to their petrogenesis. Fe/Mg inter-diffusion in clinopyroxene is ideally suited in this analysis, and we select two magmatic temperatures for analysis: 1200 °C (log$_{10}$ $D^{Mg/Fe}$ inter-diffusion = −18 m$^2$/s)[32], and 1150 °C (log$_{10}$ $D^{Mg/Fe}$ inter-diffusion = −19.4 m$^2$/s)[33]. The thinnest Fe and Mg lamella observed in clinopyroxene is about 15 μm thick (see Supplementary Fig. 6). The preservation of Fe and Mg lamellae in clinopyroxene suggest magmatic residence times of <45 years at a temperature of 1150 °C to <2 years at 1200 °C. Although there are uncertainties in the assumed temperature and in the experimental values of Fe/Mg inter-diffusion in clinopyroxene, we argue these estimates are accurate to within an order of magnitude, and potentially within 20–30%, based on experimentally determined activation energies and reasonable estimates for magma temperatures as constrained by phase equilibria[34]. The elemental concentration field observed in the clinopyroxenes exhibit an irregular zoning pattern, and along with the $^{87}$Sr/$^{86}$Sr zoning in plagioclase from Edwards et al.[16], are suggestive of at least one magma mixing event. The preservation of elemental zoning in clinopyroxene suggests that the eruption occurred relatively soon after magma mixing. The short timescales between mixing of geochemically distinct melts and the eruption of the mixture was sufficiently short for preservation of the heterogeneous $^{87}$Sr/$^{86}$Sr and $^{143}$Nd/$^{144}$Nd in the magmatic clinopyroxenes. Hence, the observed isotopic heterogeneity reflects the heterogeneity represented by the two magmas that contributed to the mixture.

**Thermodynamic modeling of the evolution ALIA-D115 lavas.** Using the two endmember compositions described above—the mafic, low $^{87}$Sr/$^{86}$Sr and the trachytic, high $^{87}$Sr/$^{86}$Sr endmembers—thermodynamically constrained mixing simulations using MELTS[24–26] and the MCS[27] at ~0.2 GPa best capture the major element and isotope characteristics of the Samoan lavas (see Supplementary Figs. 2 and 3) and validate the mixing endmembers proposed in this study. In addition, the mixing calculations approximately reproduce the dominant mineralogy of clinopyroxene (Mg#$_{71–80}$; natural clinopyroxene range from Mg#$_{71–82}$ (ref. [8]); see Supplementary Table 1) and plagioclase (~An$_{45}$; natural plagioclase range from An$_{50–60}$ (ref. [16])) as observed in the ALIA-D115-18 lava (see "Methods" for model parameterizations). It is important to note that the MCS mixing lines plotted in Supplementary Figs. 2 and 3 (plotted in gray) are not expected to follow the same path in compositional space as the closed-system binary mixing models, nor end at the "EM2"-derived endmember (see "Methods"). This is because the former tracks the predicted melt composition whereas the latter tracks the bulk composition of the mixture. It is expected that mixing of two (or more) magmas would produce crystal populations with heterogeneous major, trace, and $^{87}$Sr/$^{86}$Sr and $^{143}$Nd/$^{144}$Nd compositions since the hybridization (complete mixing or homogenization of two or more magmas) of the mixing endmember magmas is not instantaneous and most likely incomplete, even after decades of magma co-mingling[13] (where co-mingling refers to incomplete mixing between two or more magmas producing discrete compositional bands in the actively mixing magmas).

**Origin of EM2 and derived melts.** The petrogenetic modeling presented here shows that a silicic endmember EM2 magma is

consistent with the geochemical array formed by the ALIA-D115 lavas from Samoa. An important outcome of this effort is the determination that the extreme EM2 magma endmember in Samoa is trachytic in composition. The origin of the EM2 endmember is still an important question to be answered. Previous work has shown that even though EM2 has high $^{87}Sr/^{86}Sr$ and EM2 lavas at Samoa are relatively evolved, EM2 is not the result of shallow contamination of crustal material or sediments. Oceanic crust has insufficiently radiogenic $^{87}Sr/^{86}Sr$ to produce the $^{87}Sr/^{86}Sr$ observed in the ALIA-D115 lavas, and furthermore, Pb isotopic compositions of modern marine sediments from the Samoan region compared with the ALIA-D115 lavas form a non-overlapping, diverging trend such that the highest $^{87}Sr/^{86}Sr$ lavas are furthest from the marine sediment field[7,16]. Supplementary Fig. 8 shows that Nb/Th of the ALIA-D115 lavas decreases with increasing $^{87}Sr/^{86}Sr$, consistent with magma mixing between a mafic, low $^{87}Sr/^{86}Sr$ (high Nb/Th) source and one that has high $^{87}Sr/^{86}Sr$ and low Nb/Th ratios (Nb/Th = 1.8 for the low MgO EM2-derived endmember of this study) similar to upper continental crust (Nb/Th = 1.1 (ref. [35])) or average sediments (Nb/Th = 1.2 (ref. [36])). However, as stated above, neither of which are modern materials, but rather ancient subducted materials.

While the presence of this trachytic endmember (with $^{87}Sr/^{86}Sr$ up to 0.73158) is inferred from the mixing calculations presented in this study, it has not (yet) been found in pure form in lavas examined from the ALIA-D115 dataset. However, geochemical interrogation of the highest $^{87}Sr/^{86}Sr$ Samoan lavas—ALIA-D115-18 and ALIA-D115-21 (with whole rock $^{87}Sr/^{86}Sr$ of 0.718592 and 0.720469, respectively)[7]—reveals that single clinopyroxene crystals ($^{87}Sr/^{86}Sr$ up to 0.723888; this study, Supplementary Table 1) and zones in plagioclase ($^{87}Sr/^{86}Sr$ up to 0.7224)[16] sample even more extreme EM compositions than the host whole rocks and trend toward the endmember EM2 melt calculated here. More intriguingly, reports of high-SiO$_2$ (65 to 69 wt%) trachytic melt inclusions[37] hosted in the same ALIA-D115-18 clinopyroxene grains examined in this study raise the possibility that the EM2 endmember may be preserved in relatively pure form as glassy melt inclusions in the clinopyroxenes from a high $^{87}Sr/^{86}Sr$ lava (see Supplementary Table 2 for example melt inclusion compositions). Unfortunately, these melt inclusions are too small for analysis of $^{87}Sr/^{86}Sr$. Nonetheless, the presence of trachytic melts representing the EM2 endmember composition raises important questions about the origin of melts sampling this source. The fact that the model presented here indicates ~20% of a mafic magma mixed with ~80% trachytic EM2 endmember magma suggests that the potential conditions of mixing of silicic and mafic magmas may be due to an initially isolated silicic magma body that was recharged by a more mafic magma, where clinopyroxene and plagioclase crystallization (and melt inclusion entrapment) occurs during the mixing process. This scenario is consistent with the highly variable nature of $^{87}Sr/^{86}Sr$ observed in the clinopyroxene and plagioclase crystals. The question still remains, what is the origin of these silicic, trachytic melts?

Previous studies provide evidence of silicic melts in the mantle derived by various processes, such as low-degree partial melting of anomalous mantle, reaction between CO$_2$-rich fluids/melts or basaltic to silicic melts and peridotite, or as the result of extensive fractional crystallization[5,38–48]. One hypothesis tested in this study is partial melting of geochemically enriched (with high $^{87}Sr/^{86}Sr$) mantle compositions followed by crystal fractionation to produce trachytic melts. Considering a scenario in which ancient subducted sediment has infiltrated and mixed with ambient peridotite, partial melting of this source could give rise to silicic melts with radiogenic $^{87}Sr/^{86}Sr$. Results (described in detail

in the Supplementary text) show that partial melts (at a range of pressures) of various mixed sediment+peridotite sources are consistently basanitic (high-degree partial melts) to phonolitic (low-degree melts) and subsequent fractionation of these melts does not produce trachytes. However, we acknowledge that we have not undertaken an exhaustive search of parameter space (e.g., different sediment compositions, different combinations of sediment±recycled oceanic crust±peridotite, various H$_2$O and CO$_2$ source concentrations, and a full range of fO$_2$, pressures, and temperatures), and future work will focus on melting and fractional crystallization processes in the petrogenesis of compositions similar to the EM2 trachytes presented here.

A scenario in which trachytic melts are formed directly through partial melting of a sediment-like source and subsequently mixed with more mafic Samoan melts to form the ALIA-D115 mixing trend is also possible. However, silicic melts are highly reactive with ultramafic bulk compositions and a transport scenario in which large volumes of this melt travel through 10s of kilometers of ocean mantle lithosphere to shallow magma chambers and remain isolated, requires further study (see Adams et al.[49], for a further discussion).

Trachytic melts have been shown to be common products of metasomatic processes such as reaction of ambient peridotite mantle with CO$_2$-rich fluids or silicic melts[4,5,43,47,48,50–52]. Many workers have also suggested that partial melting of mantle already modified by metasomatic processes can produce disequilibrium silicic liquids that might vary substantially in composition depending on the source[5,47,48,53]. Evidence of carbonatitic and silicic melt metasomatism in Samoan xenoliths[4,5,54,55] exists, and a trachytic glass found in one highly carbonatite metasomatized peridotite mantle xenolith from Savai'i Island—the same island from which the lavas of this study are derived—exhibits radiogenic $^{87}Sr/^{86}Sr$, but is still much lower than the predicted $^{87}Sr/^{86}Sr$ of this study for the EM2-derived endmember. Further, as noted earlier, it remains to be determined how low melt fractions of trachyte melt observed in a xenolith can be scaled up to larger volumes and additionally, it is not clear how to transport this trachyte melt through the mantle, from the melt source to the magma mixing region. This type of petrogenetic mechanism, which invokes carbonatite metasomatism of peridotite to generate the trachytic melt source, may be a robust avenue for future research into the origin of EM2 that could be tested using thermodynamic phase equilibria modeling and/or experimental analysis (see Adams et al.[49], for further discussion).

The origin of EM2 and why it manifests in extreme form in Samoa remain important problems: EM2 is identified in lavas from Society and Marquesas Islands[17,56], for example, but the magnitude of the geochemical enrichment at these localities is less pronounced than at Samoa. Nonetheless, the new radiogenic isotopic compositions derived in this study support models advocating for a large fraction of recycled terrigenous sediment into the Samoan plume[7,16]. This further suggests that, since EM mantle domains have been found to correspond geographically with the LLSVPs[57,58], sediment-rich domains can survive in the LLSVPs over geologic timescales to be sampled by the Samoan plume.

## Methods
### TIMS analysis
*Sample preparation.* Seventeen clinopyroxene grains lacking visible attached groundmass or surface alteration were plucked from indium mounts. The clinopyroxenes were then leached in concentrated HNO$_3$ at 90 °C for 2 min and then washed once with milli-Q (18.2 × 10$^6$ Ω cm) water; this process was repeated twice to ensure that any indium remaining on the grains was removed. The clinopyroxenes were then leached in 6 N HCl at 90 °C for 2 min to remove any remaining surface contamination, then they were rinsed three times with milli-Q water. Seventeen clinopyroxene samples, together with two aliquots of USGS Reference

Material BCR2 hosting 10.8 ng Sr and 0.88 ng Nd (the first aliquot), and 5.6 ng Sr (the second aliquot, but Nd was not analyzed), respectively, were individually transferred to separate vials containing 150 μl of concentrated $HNO_3$ and 225 μl of concentrated HF for dissolution. The samples were then set on a 120 °C hotplate for 2 days for dissolution. All clinopyroxenes and BCR2 aliquots were spiked with [84]Sr and [150]Nd for determination of the total amount of analyte in the sample by isotope dilution. Samples were then dried down and later brought up in a 500 μl concentrated $HNO_3$ solution and placed on a 120 °C hotplate to flux for 24 h to eliminate fluorides. Following dry-down, the samples were brought up in 1 ml of 3 N $HNO_3$ to load on to columns for chemical separation of Sr and Nd; column chemistry follows Koornneef et al.[59]. Total procedural blanks were processed together with sample clinopyroxenes and the BCR2 through all steps of sample dissolution (beginning with dissolution in $HNO_3$ and HF), column separations, and mass spectrometry. Total procedural blanks (including all stages of wet chemistry, e.g., sample dissolution, column chemistry, etc., and loading on TIMS filaments) varied from 22 to 60 pg for Sr (and averaged 45 pg in one session and 37 pg in the other), and 2.2 to 2.7 pg for Nd. The Sr and Nd blanks are dwarfed by the total amount of Sr (~5.6 to 58.7 ng) and Nd (~0.88 to 25.6 ng), respectively, in each clinopyroxene grain and the two BCR2 aliquots. The [Sr]$_{sample}$/[Sr]$_{blank}$ and [Nd]$_{sample}$/[Nd]$_{blank}$ ratios for each sample are provided in Supplementary Table 1. Thus, while blanks corrections are applied to the samples and the BCR2 aliquots (assuming a lab blank [87]Sr/[86]Sr of 0.711 and [143]Nd/[144]Nd blank of 0.5118, obtained by pooling multiple blanks), blank corrections to the [87]Sr/[86]Sr and [143]Nd/[144]Nd ratios are negligible.

*[87]Sr/[86]Sr and [143]Nd/[144]Nd isotope analyses.* All of the clinopyroxene grains were processed through the same batch of column chemistry with a BCR2 and two total procedural blanks; two batches of chemistry were required for Sr isotopes (so two BCR2 analyses for Sr are reported) and one batch of chemistry was required for Nd isotopes (so one BCR2 analysis for Nd is reported). The clinopyroxene grains and the BCR2 were analyzed for [87]Sr/[86]Sr and [143]Nd/[144]Nd over two analytical sessions. The separated, dried Sr samples were brought up in 1 μl of $HNO_3$ and each loaded onto outgassed, zone-refined rhenium (99.999%) filaments (H Cross, USA) along with a 1 μl TaCl$_5$ emitter solution. For each analytical session, two total procedural blanks were also loaded onto rhenium filaments in addition to a BCR2, all of which passed through the same column chemistry as the clinopyroxene and BCR2 unknowns. In addition, eight filaments with 1 ng NBS987 were analyzed together with the clinopyroxenes and BCR2 aliquots: average [87]Sr/[86]Sr = 0.710262 ± 0.000076 (2 SD, $N = 8$). All samples were analyzed by static Faraday collection without amplifier rotation on UCSB's Triton Plus employing $10^{11}$ Ω amplifiers and using a 3.3 picoamp gainboard, and gains were measured every barrel. Samples were corrected for the offset between the preferred (0.710240) and the average measured NBS987 [87]Sr/[86]Sr from the same analytical session (i.e., the same barrel). Sr isotopes were corrected for mass bias assuming an exponential law and using canonical [86]Sr/[88]Sr ratio of 0.1194, and isobaric interferences from Rb were corrected by monitoring mass 85, but changes to the [87]Sr/[86]Sr ratios due to this correction were negligible.

Only nine of the above 17 clinopyroxene grains were analyzed for [143]Nd/[144]Nd; these were selected after [87]Sr/[86]Sr was analyzed and were chosen because they spanned the range of measured [87]Sr/[86]Sr values. The dried down, separated Nd samples were brought up in 4 μl of 1 M $HNO_3$ and loaded onto outgassed, zone-refined (99.999%, H Cross, USA) double rhenium filaments. Two blanks, six 1 ng JNDi's (average 0.512106 ± 0.000033, 2 SD), one 500 ng JNDi (0.512112 ± 0.000011, 2SE), and a BCR2 (run through the columns) were also loaded onto rhenium filaments and run during the same analytical session (i.e., the same barrel) as the clinopyroxenes. Samples were corrected for the offset between the preferred (0.512099)[60] and the average measured JNdi [143]Nd/[144]Nd from the same analytical session. All samples were analyzed by static Faraday collection without amplifier rotation on UCSB's Triton Plus employing $10^{13}$ Ω amplifiers and using a 3.3 picoamp gainboard (gains are measured once, at the start of the new barrel). Nd isotopes were corrected for mass bias assuming an exponential law and using a canonical [146]Nd/[144]Nd ratio of 0.7219. Isobaric interferences from Sm were corrected by monitoring mass 147, but changes to the [143]Nd/[144]Nd ratios due to this correction were negligible.

At UCSB, the long-term reproducibility (up to and including this study) of [87]Sr/[86]Sr for 1 ng and 500 ng loads of NBS987 by static multicollection (without amplifier rotation) using the same methods as this study are 0.710248 ± 0.000061 (2 SD, $N = 69$) and 0.710244 ± 0.000014 (2 SD, $N = 39$), respectively. For BCR2, multiple runs of aliquots hosting 5.6 to 10.8 ng Sr that were spiked and processed through column chemistry and mass spectrometry (following the same methods as samples here) yield an average [87]Sr/[86]Sr of 0.705018 ± 0.000065 ($N = 9$), which is in line with the [87]Sr/[86]Sr of the two BCR2 aliquots analyzed in this study: the 10.8 ng aliquot [87]Sr/[86]Sr = 0.704974 ± 0.000018 (2SE), and the 5.6 ng aliquot [87]Sr/[86]Sr = 0.705027 ± 0.000045 (2SE). These values are comparable to an average [87]Sr/[86]Sr of 0.705005 ± 0.000010 for BCR2 reported by Weis et al.[61] (following normalization to the same NBS987 value used here). For [143]Nd/[144]Nd, the long-term reproducibility of 1 ng loads of JNdi by static multi-collection (without amplifier rotation) using the same methods as this study (i.e., $10^{13}$ Ω amplifiers) is 0.512104 ± 0.000030 (2 SD, $N = 27$). For BCR2, 0.50 to 0.88 ng Nd aliquots spiked and processed through column chemistry and mass spectrometry (processed

together with the samples here) yield an average [143]Nd/[144]Nd value of 0.512618 ± 0.000023 ($N = 4$); the [143]Nd/[144]Nd of the BCR2 aliquot analyzed here (0.88 ng Nd) was 0.512634 ± 0.000032 (2 SE). This is comparable to an average [143]Nd/[144]Nd value of 0.512621 ± 0.000012 for BCR2 reported by Weis et al.[61] (following normalization to the same JNdi value used here, and the La Jolla to JNdi conversion of Tanaka et al.[62]).

**Clinopyroxene major and trace element analysis.** Clinopyroxene major elements reported in Supplementary Table 1 were collected by the electron probe microanalyzer (EPMA) at the University of California Santa Barbara. An accelerating voltage of 15 kV was used with a 20-nA beam and a 2-μm spot size. Each oxide is the average of many 2-μm pixels across a fully quantitative x-ray map performed on each grain. All data for each element map were filtered to remove any inclusions within the clinopyroxenes prior to averaging. Low total measurements from individual pixels (<90 wt%) were also filtered out of the dataset prior to averaging.

Clinopyroxene trace element concentrations were collected at the University of California Santa Barbara (Supplementary Table 1) by LA-ICP-MS using a Photon Machines Excite 193 Excimer laser coupled to an Agilent 7700 quadrupole ICP-MS. A 15-μm spot diameter was used. Unknowns were corrected relative to reference material NIST612 analyzed every 8–10 unknown analyses; analyses and the 2RSD on each element can be found in Supplementary Table 4. Each trace element measurement reported in Supplementary Table 1 is an average of three to four spot analyses spread across each clinopyroxene grain.

**Residence times from diffusion.** To obtain an estimate of the residence time, defined as the time interval between crystal growth and lava eruption (quenching), of the clinopyroxene porphyrocrysts, the diffusive characteristic of lamellar compositional bands is considered. Examination of compositional profiles revealed that thin lamella of locally higher Mg/Fe were preserved (Supplementary Fig. 6). Based on Fe–Mg inter-diffusion, we used the thickness of these lamella to obtain an estimate of the maximum residence time of the crystal. That is, if diffusion was active for a period of time greater than the residence, the compositional anomaly would have been erased. In this one-dimensional diffusion model, a lamella of greater Mg/Fe than its surroundings initially at composition $C_0$ occupies a region between $x = -b$ and $x = +b$. The composition outside the lamella is $C_1$ at all times. The non-dimensional differential equation governing the decay of the lamella compositional anomaly is

$$\frac{\partial \widehat{C}}{\partial \widehat{t}} = D \frac{\partial^2 \widehat{C}}{\partial \widehat{x}^2}, \tag{1}$$

where the non-dimensional variables are defined as:

$$\widehat{C} = \frac{C_1 - C}{C_1 - C_0}, \ \widehat{x} = \frac{x}{b}, \text{ and } \widehat{t} = \frac{Dt}{b^2}$$

where $2b$ is the lamella thickness, $D$ is species-appropriate diffusion coefficient, and $t$ is the time since the start of diffusion. The initial and boundary conditions in terms of the non-dimensional parameters are:

IC: at $t = 0$, $\widehat{C} = 1$ for $-1 < \widehat{x} < 1$
BC: at $\widehat{x} = \pm 1$, $\widehat{C} = 0$ for $\widehat{t} > 0$
The solution[31] recast in dimensional variables is:

$$\frac{C_1 - C}{C_1 - C_0} = 2 \sum_{n=0}^{\infty} \frac{-1^n}{(n+\frac{1}{2})\pi} \exp\left[-\left(n+\frac{1}{2}\right)^2 \pi^2 \frac{Dt}{b^2}\right] \cos\left(n+\frac{1}{2}\right)\frac{\pi x}{b}. \tag{2}$$

Based on this solution, we computed the time necessary to eradicate 90% of the compositional anomaly at the centerline of the lamella (see ref. [18], pg. 101 for graphical solution) for the thinnest lamella in the clinopyroxenes: ~15 μm ($b = 7.5$ μm). Diffusion calculations were performed at 1150 °C (log$_{10}$ $D^{Mg/Fe \text{ inter-diffusion}} = -19.4$ m$^2$/s)[33] and 1200 °C (log$_{10}$ $D^{Mg/Fe \text{ inter-diffusion}} = -18$ m$^2$/s)[32]. The effect of varying temperature ±50 K is modest, increasing or decreasing residence times by about 20–30%. The greatest uncertainty is in the laboratory determination of diffusivities.

**Retrieval of magma mixing endmembers.** The principles of magma mixing are well established and have been used for decades to decipher magma mixing from assimilation and/or fractionation processes in natural magmatic systems[23,63–66]. The Samoan ALIA whole rock lavas discussed in this study are very likely a result of magma mixing as noted by examination of Fig. 3. The mixing trends can be used to estimate the composition of the mixing endmembers given the MgO content of the silicic (S; high [87]Sr/[86]Sr) and mafic (M, low [87]Sr/[86]Sr) mixing endmembers. Based on binary mixing theory[23,65,66], for any two chemical species (major oxides, trace elements, isotopes), the mixing equations may be recast in the ratio-reciprocal form

$$\frac{C_2^H}{C_1^H} = A + \frac{B}{C_1^H} \tag{3}$$

where $C_1^H$ and $C_2^H$ are concentrations of chemical species 1 and 2 in the mixed (hybrid) magma, respectively, and $A$ and $B$ are constants related to the chemical

species concentrations in endmembers M and S. The constants A and B in Eq. (3) are defined from mixing theory to be:

$$A = \frac{C_2^M - C_2^S}{C_1^M - C_1^S} \qquad (4)$$

and

$$B = \frac{(C_1^M C_2^S - C_1^S C_2^M)}{C_1^M - C_1^S} \qquad (5)$$

where $C_1^M$ and $C_2^M$ are concentrations of chemical species 1 and 2 in the mafic (M) endmember and $C_1^S$ and $C_2^S$ are the concentrations in the silicic (S) endmember. The linear regression of the petrochemical data for the ALIA-D115 lavas gives the intercept and slope that corresponds to A and B. A and B are two equations with 4 unknowns, thus if the concentration of chemical species 1 is known, or can be confidently estimated, in both endmembers, then the concentration of chemical species 2 in both endmembers can be calculated by simple rearrangement of Eqs. (4) and (5). The quality of the computed parameters depends on the quality of the fit of the data to Eq. (3). We have chosen to initially fix values of MgO in both M and S because there is relatively little variability in MgO concentrations in both endmembers based on the array defined by the ALIA-D115 lavas. In other words, a plot of Samoan lavas in $SiO_2$ versus MgO space (see Supplementary Fig. 2), shows the linearity expected for mixing, as the ALIA-D115 lavas form a trend with the M endmember sitting in the cloud of data points of "other Samoan shield lavas" and the S endmember extending toward 0 wt% MgO, thus, the MgO in the S mixing endmember must lie between the value of the most evolved and high $^{87}Sr/^{86}Sr$ ALIA lava (ALIA-D115-21) and 0 wt% MgO (see Supplementary Fig. 2). The M endmember is assigned a value of 10 wt% MgO (similar to the MgO composition one would get if all "other Samoan shield lavas" with less than 50 wt% $SiO_2$ were averaged) whereas for the S endmember, we have chosen to perform the mixing calculations for three different values of MgO: 3.8 wt% (from ALIA115-21), 2 wt%, and at the low end of the spectrum, 0.5 wt% (which is very close to the MgO content of clinopyroxene-hosted trachytic melt inclusions; see Supplementary Table 2) to cover the possible MgO contents. The method allows one to calculate the oxide compositions of M and S endmembers for each of the three assumed values of MgO in S, given a constant value in M. In detail, we used the A and B values from the regression of expressions of the following form: $C_i/MgO$ versus $1/MgO$ space, where $C_i$ are the following chemical species: $SiO_2$, $Al_2O_3$, $Na_2O$, $K_2O$, and Nd (see Supplementary Fig. 4).

From these plots, we computed an estimate of the M and S bulk compositions for $SiO_2$, $Al_2O_3$, $Na_2O$, $K_2O$, and Nd (see Supplementary Fig. 4). As an example, with a linear least squares regression equation describing data in $K_2O/MgO$ versus $1/MgO$ space, with some re-arranging of Eqs. (4) and (5) we can solve for the concentration of $K_2O$ in both the M and S endmembers given our estimated initial values of MgO for each endmember as known values. For the chemical species other than $SiO_2$, $Al_2O_3$, $Na_2O$, $K_2O$, and Nd, we used the computed $K_2O$ concentration as the new denominator value (instead of MgO) to calculate the concentrations of $TiO_2$, FeO, CaO, MnO, $P_2O_5$, and Sr. The reason for this is that plots of these elements with $K_2O$ as chemical species 1 in Eq. (3) showed much better linearity compared to equivalent plots using MgO in the denominator—we need the best correlations possible to get a best estimate on the two endmember compositions.

According to mixing theory, "companion plots" to the ratio–ratio plot $^{87}Sr/^{86}Sr$ versus $^{143}Nd/^{144}Nd$ should give linear trends and are of the form $^{87}Sr/^{86}Sr$ versus Nd/Sr and $^{143}Nd/^{144}Nd$ versus Sr/Nd (Supplementary Fig. 5). Thus, these plots are used again with ordinary least squares regression techniques and, given the inferred Sr and Nd concentrations in each endmember derived in the manner described above, we extract $^{87}Sr/^{86}Sr$ and $^{143}Nd/^{144}Nd$ in both endmembers (see Supplementary Table 2) using again, Eqs. (4) and (5). The last step of the process is to use the calculated $^{87}Sr/^{86}Sr$ and $^{143}Nd/^{144}Nd$ and Sr and Nd concentrations to obtain a self-consistent mixing curve to the ALIA-D115 lavas, in addition to the new clinopyroxene data of this study, in a plot of $^{87}Sr/^{86}Sr$ versus $^{143}Nd/^{144}Nd$.

The values of the reconstructed major oxide, Sr and Nd concentrations, and isotopic ratios for the mafic and silicic magma mixing endmembers for each of the three assumed MgO contents of the R endmember are given in Supplementary Table 2. The three high $^{87}Sr/^{86}Sr$ endmember reconstructed compositions (using the three different choices of MgO for the endmember) lie either in the trachyte (two of three) or trachyandesite (one of three) field on the total alkali versus silica diagram.

Lastly, it is important to note that although some correlations in the ratio–ratio plots (see Supplementary Fig. 4) are not as highly correlated as others, likely related to crystal fractionation effects, it does not actually matter significantly what elements are used in the denominator in these calculations (we have chosen to use MgO and $K_2O$, because these are the most well constrained by the data), the derived endmember compositions will be quite similar. The values change very slightly based on how good the correlations are. As an example, we used the ratio of $Sr/K_2O$ vs. $1/K_2O$ to derive Sr concentrations in the two endmembers with a correlation coefficient of 0.96; the resultant Sr concentration in the mafic endmember was 712 p.p.m. and 380 p.p.m. in the silicic endmember. If we had instead performed the same calculation with $Sr/MgO$ versus $1/MgO$, the correlation coefficient is 0.85 and the Sr concentration in the mafic endmember

would be 726 p.p.m. and the silicic endmember would be 360 p.p.m. These differences are comparable to analytical uncertainties for real measurements.

**MCS mixing modeling.** Using the calculated mafic (M; low $^{87}Sr/^{86}Sr$) and silicic (S; high $^{87}Sr/^{86}Sr$) mixing endmembers from the above procedure, MELTS[24,25] and the MCS[27] were used to simultaneously model the effects of magmatic recharge and fractional crystallization processes on Samoan lavas (RFC in MCS jargon). The metric used to gauge model acceptability was how well the phase equilibria and geochemistry matches the petrology and petrochemistry of the observed Samoa ALIA-D115 lava suite. In the magma mixing calculations, the M endmember composition was the host magma and the S endmember composition, based upon an assumed MgO content of 0.5 wt% (see Supplementary Table 2 for details), was the recharge magma. The $H_2O$ content of both M and S endmembers was set at 0.1 wt% along the QFM (quartz-magnetite-fayalite) buffer establishing the ferric to ferrous iron ratio in each endmember. Since the porphyrocrystic clinopyroxenes of this study are in isotopic disequilibrium with the whole rock, clinopyroxene-liquid thermobarometry is not meaningful. Because an independent pressure estimate could not be made, MCS simulations were performed in the range 0.1 to 1 GPa. Upon mixing at low pressures (<0.5 GPa), the stable fractionating mineral assemblage consists of olivine, clinopyroxene, plagioclase, and spinel. Simulations were run for three cases where the high $^{87}Sr/^{86}Sr$ recharge magma (S endmember) was initially either: (1) 100% liquid, (2) possessed a crystallinity of about 15% (15% $An_{22}Ab_{63}Or_{15}$ anorthoclase feldspar, modally), and (3) possessed a crystallinity of 60% (70% $An_8Ab_{48}Or_{44}$ alkali feldspar, 23% $An_{20}Ab_{60}Or_{20}$ plagioclase feldspar, 5% spinel, and 2% orthopyroxene, modally) upon addition to the host magma (M endmember). At constant pressure, the difference between these three initial recharge magma thermodynamic states is that as the crystallinity of the recharge magma increases the initial anorthite content and Mg# of the plagioclase and clinopyroxene, respectively, that crystallize from the host magma increases upon the initiation of mixing. As an example, at 0.2 GPa, when the recharge magma was 100% liquid, plagioclase ($An_{45}$) was stabilized at 1107 °C whereas when the recharge magma was 60% crystalline, $An_{50}$ plagioclase was stabilized at ~1119 °C. In the case of the ALIA-D115 lavas, the latter case more closely corresponds to observed plagioclase within the lavas (range $An_{50-60}$)[16]. With increasing pressure, the first plagioclase to crystallize from the magma decreased in An content and there was increased stabilization of orthopyroxene in the assemblage over clinopyroxene, which is not observed in the ALIA-D115 lava suite. Thus, MCS simulations favor lower pressures of magma mixing and a partly crystalline recharge magma at the time of mixing. Overall, the modal amounts and phase compositions of the calculations agree quite well with the observed mineralogy. It is important to note that the MCS mixing lines plotted in Supplementary Figs. 2 and 3 are the evolutionary paths of the *melt* as a result of magma mixing and crystallization (crystals are removed from the system as soon as they form), so the bulk composition is not conserved and thus, one would not expect the mixing lines to end at the EM2-derived endmember, much different than the conservative mixing line based on closed system binary mixing. In addition, the MCS models are thermodynamically constrained models and so the terminus of the mixing line is dictated by where the simulation terminated. In any regard, the ALIA-D115 lavas can be explained reasonably well given the ubiquitous geological uncertainties in any modeling exercise.

## Data availability
All data generated in this study are available in the Supplementary data files of this article and can also be found on the IEDA EarthChem online database at the following link: https://doi.org/10.26022/IEDA/111769. Source data for figures can be found within the Supplementary Data files.

## Code availability
The current Magma Chamber Simulator (MCS) source code, details on how to operate MCS, and numerous applied examples can be found at https://mcs.geol.ucsb.edu/. Also see Bohrson et al.[27], Bohrson et al.[67], and Heinonen et al.[68] for more details on MCS operations and applied examples.

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

## Acknowledgements
J.V.A. acknowledges the GSA graduate student research grant that partially funded the TIMS isotope analysis. M.G.J. acknowledges NSF grants OCE-1736984, EAR-1429648, EAR-1900652. We thank Stan Hart and Hubert Staudigel for contributing sample material.

## Author contributions
J.V.A. performed the theoretical modeling, mixing calculations, much of the analytical work, and wrote the manuscript. M.G.J. provided the samples and conceived the project idea. F.J.S. helped formulate the binary mixing and diffusion calculations. M.G.J. and F.J.S. contributed significant additions and edits to the manuscript. Clinopyroxene sample preparation and TIMS analyses were performed by J.V.A., A.A.P. and B.L.B. Clinopyroxene EPMA major element maps and data analysis was performed by G.S. and J.V.A. Clinopyroxene trace element analyses were performed by J.V.A. with data reduction and analysis help by J.M.C. All authors participated in discussion and/or interpretation of results and gave valuable feedback on the manuscript.

## Competing interests
The authors declare no competing interests.
