## [Peer Review File · Nature Communications]

REVIEWER COMMENTS

Reviewer #1 (Remarks to the Author):

This study provides new Sr and Nd isotope data for individual crystals of clinopyroxene, taken from one lava that was erupted at the Samoa hotspot. The lava chosen for study has the second most extreme whole rock composition for Sr-Nd isotopes ever sampled at Samoa. Samoa is the type locality that is influenced by the extreme EM2 (Enriched Mantle type 2) component that is used to help describe the global variability of oceanic basalts. The new paired Sr-Nd results are the first of their kind for single crystals from this ocean island, and they display a remarkably wide range that encompasses about one-third of the variation in ocean islands globally. The variability within this single lava, along with the supporting chemical information, supports an origin by mixing of magmas and not of mantle sources. In light of that, the new results are novel in that they have been used within a petrologic modeling framework, to provide a much deeper insight than was previously understood, about the origin of the large isotopic variability seen in Samoan lavas.

The paper breaks new ground in the field of mantle geochemistry even after decades of study and a significant number of earlier studies on Samoa in particular. The study is well conceived, and the analytical data are of state-of-the-art quality for these types of samples. The paper will be of interest to readers beyond those specializing in mantle geochemistry, to include igneous petrologists and geodynamicists, and so should be of broad appeal to the readership of *Nature Communications*. For the most part the paper is all ready for publication, and I recommend acceptance after the authors address the points described below.

General comments:

First, I should say that I am not an experienced user of MELTS or the Magma Chamber Simulator. I therefore cannot quantitatively evaluate the results described in supplementary Table 3a or 3b, and can only state that the petrologic modeling and its results seem internally consistent and accurate based on what I know from reading previous work.

1. The paper would be improved somewhat if the very end provided a bit more discussion about what the new results imply (if anything) about the extent / occurrence of EM2-derived compositions as a mantle isotopic component globally. Although extensive discussion is not appropriate for this paper's format, the end of the paper left me wondering what the authors were thinking are the new questions that have emerged (Line 302 simply states - "the presence of trachytic melts representing the EM2 end-member composition raises important questions about the origin of melts sampling this source"). Recognizing that trachytes are not commonly thought of as directly derived mantle melts, and following on from the last parts of the discussion in the supplementary material, it seems to me that more should be said here. The authors also seem to (perhaps) imply the new results hint somehow at how long recycled sediment might reside in the mantle source of OIBs that have an EM2 signature in this end paragraph, but I am unclear how that would be any different from what has already been discussed in the literature for a few decades now along the lines of mantle component origin/residence. The connection of EM domains to the LLSVPs seems to try to bring this into the modern picture of mantle dynamics (line 307). But it seems this last paragraph has mixed a few ideas together, and maybe missed an opportunity for discussing how the new results and modeling should influence our thinking about the regional or global origin and extent of the EM2 composition.

2. Following on from comment 1, it seems appropriate to point out somewhere in the paper the occurrence of somewhat unusual, high silica melts at other ocean islands that in some ways resemble the trachytic end-member melts proposed in this study. Examples include Réunion (Ludden (1976) *Canadian Mineralogist* 16, pp. 265-276), La Palma, Canary Islands (Wulff-Peterson et al, (1999) *Contrib. Mineral. Petrol.* 137, pp. 59-72), Kerguelen (Schiano et al. 1994, *EPSL* 123, 167-178), and Grande Comore (Schiano et al. 1998, *EPSL* 160, 537-550). There are probably other cases of which I am unaware. One of the above occurrences (Réunion) may have originated

by extreme differentiation within the crust; however, in most cases above the compositions are found as melt inclusions in mafic igneous phases, and have been suggested to ultimately originate by various processes that may include the formation of immiscible alkali-rich liquids in basaltic melts (Réunion), by reaction between host peridotite and infiltrating basaltic melts (La Palma), by metasomatism from exotic (non-peridotitic) melts (Kerguelen), or as the very earliest melts derived from spinel peridotite (Grande Comore). I am not sure how much more can be made of this that stems from the present study (some ideas are broadly mentioned with regard to Samoa in the supplementary discussion of model scenarios 2 and 3, but other island occurrences are not really mentioned there). It seems the presence of these types of melts may be more widespread than recognized and should be mentioned.

3. There are a number of terms used in the paper that are common to igneous geochemists, that made it a bit of scramble for me to look around in the paper (sometimes ahead) to try to really understand what precisely the authors meant. The terms include "evolved", "mixing", "hybridization" and "co-mingling" (these are most obvious in the discussion that begins on line 282). Is co-mingling just inefficient mixing, or no mixing at all, just their juxtaposition? How does hybridization fit in the order of these things? An extensive explanation is not needed, but I do think it would help if the authors stated more precisely what they envision these terms mean when they first use some of them.

Minor comments:

1. Abstract. It is stated in the paper that "a direct comparison of isotopic heterogeneity observed in clinopyroxene and plagioclase within a single hand specimen provides new constraints on the source of isotopic heterogeneity..." (line 96-97; with further important details described on lines 163-179) but this cpx-plag comparison is not mentioned in the abstract, and instead it only mentions clinopyroxene. Line 50 would have been more clear (to me at least) if the terms evolved = "evolved (trachytic)" or simply "silicic" and less evolved = "less evolved (basanitic)" or simply "mafic". Also, the last sentence of the abstract would be better used to describe what the important new constraint(s) is(are) based on the new observations rather than the general statement that is currently provided.

2. Line 147 (caption to figure 1) should add that the 9 of 17 crystals analyzed for Nd isotopes encompass the full range of $^{87}\text{Sr}/^{86}\text{Sr}$.

3. Line 260. I was confused by the statement of how "no coherent zoning pattern" in clinopyroxene is evidence for at least one magma mixing event. Based on Fig. S6, I think it would be better described in this sentence (L260) as an "irregular zoning pattern" (at least that seems less confusing to me).

4. I found the discussion from line 431 to 443 rather essential to my acceptance that the modeling seemed internally consistent with a reading of the rest of the paper. In particular, the observation that the MgO in each end-member is confined to a rather narrow range of values based on the lava compositions themselves, and the linearity of the SiO₂-MgO mixing that extends between the other island data and approaches a fictive end-member with 0% MgO. I think it would be useful to include mention of these points in the main text where the modeling is outlined rather than just the supplementary material. The use of K instead of Mg makes sense when looking at some of the other elements because it should help to reduce spurious correlation effects. However, whenever elemental ratios are plotted using a common denominator, the possibility of seeing a spurious correlation cannot always be eliminated unless other criteria are met (e.g. Jackson & Somers 1991, The spectre of spurious correlations, *Oecologia* 86, 147-151). Some of the trends seen in Fig. S4 are not as highly correlated as others (e.g., involving K₂O, MnO, FeO and P₂O₅) and some spurious effects may contribute to some of them. This does not call into question the suitability of the approach, which clearly stems from the observations of whole rock isotopes and MgO and SiO₂ as mentioned. I do think the authors might want to critically assess how much of the simple binary mixing vs. (seemingly minor) spurious effects are responsible for the different elements they are

showing in this figure. Along the same line, figure S5 shows the companion binary mixing plots (involving the elemental ratios) to the $^{87}\text{Sr}/^{86}\text{Sr}$ - $^{143}\text{Nd}/^{144}\text{Nd}$ diagram. Simple binary mixing should also be (even more) evident on plots including the reciprocal elemental ratios ($1/\text{Nd}$ and $1/\text{Sr}$), but these are not discussed in the supplement. If simple binary mixing is responsible, then these plots also allow an extreme limit to be placed on the isotope composition of the end-member where the reciprocal concentration (i.e., $1/\text{Nd}$ or $1/\text{Sr}$) goes to zero. Obviously this will be complicated by the fractional crystallization of plagioclase as discussed in the main text, but that effect also influences the other supplementary figures mentioned above that involve Sr.

5. Line 841 compares equilibrium vs. batch melting as though they are different. Maybe fractional melting (or critical melting) vs. batch melting was intended?

5. The acknowledgments include E.A.P. (not one of the authors?).

David Graham
April 21, 2020

Reviewer #3 (Remarks to the Author):

This manuscript by Adams et al., reports impressive coupled Sr-Nd isotopic data on a set of clinopyroxene phenocrysts from a dredge sample from Samoa. The major claim is that the isotopic variability results from binary mixing of a mafic and an evolved trachytic melt, the latter which is suggested to represent a EM2 component that includes old recycled sediments. The suggestion that the endmember melt has a trachytic compositions is novel.

The high-quality data are exceptional and intriguing, and their implications are of significant interest to the community. This to me justifies publishing this paper in Nature Communications. The manuscript is well-written and the model calculations are described in enough detail as to understand and reproduce them. The way the data is interpreted and discussed currently, however, can in my opinion be more robust and could do with some clarification and restructuring. I have a few more major points that I raise here, more detailed comments and minor edits can be found in the annotated manuscript.

1) The authors present coupled major and trace element and Sr-Nd isotopic data of clinopyroxenes. However, in their interpretation and mixing models they use the bulk lava compositions rather than the clinopyroxene data that is argued to record the mixing process in more detail. Can they not use the cpx major and trace element data to calculate theoretical parental melts? I can guess what could be the reason that would not work very well, but think it would be fair to address this in the paper. I assume the relations between the elemental data and the isotopes in the clinopyroxenes are less well defined or absent? Why? Can part of this be explained by the fact that the LA-ICPMS data is not representative for the bulk cpx and would it maybe have been better to do ICPMS on a aliquot of the dissolved sample?

2) The mixing model is built on relations between Sr-Nd isotope compositions and major element compositions, with MgO being an important fixing point. Major elements are known to be prone to crystallisation of major phases such as olivine, pyroxene or feldspar, so to me it would have been more robust to use trace element ratios to avoid the influence of fractionation. The model seems to work for Sr/Nd ratios, but I am curious to see if other ratios of highly incompatible elements that are commonly used as source indicators also work for the model; e.g. Ce/Pb, Nb/U, Nb/La, Th/Nb etc

3) Even though I agree that the data seems to suggest some sort of binary mixing process; I think the authors should address in more detail what the physical implications of their theoretical model are. For example the conditions of magma genesis and mixing are not addressed in detail in the text. First of all, some sentence(s) should be included to argue why it cannot be AFC in the crust (absence of CC and depth constraints?). But also, the claim for the geochemical relations to reflect

binary mixing of mantle derived melts requires how that could physically work. What are the implications for the mechanism of melt generation, mixing and extraction? How can we have had a system where 80% of trachyte melt mixed with 20% mafic melt to generate the most extreme cpx? Did the systems stay isolated until cpx crystallised? What does it infer with respect to the size and distribution of recycled components? Why do we only see the heterogeneity for this locality? Some of that discussion is now in the supplementary text, but should be expanded by using related literature (currently there is quite a large amount of self-citation) and moved to the body text. I think there is no strict word limit that would not allow that.

4) There is some important petrographical information that is either presented very late in the manuscript (trachytic melt inclusions in the cpx) or in the supplement alone (plag mineral inclusions within the cpx). Some of that data is used as an argument to support the interpretation (line 297), but I guess the petrography should rather be the fundament for the interpretations.

I hope my comments and suggestions will help to improve a revised version of the manuscript to be accepted in this journal.

Janne Koornneef

Reviewer #1 (Remarks to the Author):

This study provides new Sr and Nd isotope data for individual crystals of clinopyroxene, taken from one lava that was erupted at the Samoa hotspot. The lava chosen for study has the second most extreme whole rock composition for Sr-Nd isotopes ever sampled at Samoa. Samoa is the type locality that is influenced by the extreme EM2 (Enriched Mantle type 2) component that is used to help describe the global variability of oceanic basalts. The new paired Sr-Nd results are the first of their kind for single crystals from this ocean island, and they display a remarkably wide range that encompasses about one-third of the variation in ocean islands globally. The variability within this single lava, along with the supporting chemical information, supports an origin by mixing of magmas and not of mantle sources. In light of that, the new results are novel in that they have been used within a petrologic modeling framework, to provide a much deeper insight than was previously understood, about the origin of the large isotopic variability seen in Samoan lavas.

The paper breaks new ground in the field of mantle geochemistry even after decades of study and a significant number of earlier studies on Samoa in particular. The study is well conceived, and the analytical data are of state-of-the-art quality for these types of samples. The paper will be of interest to readers beyond those specializing in mantle geochemistry, to include igneous petrologists and geodynamicists, and so should be of broad appeal to the readership of Nature Communications. For the most part the paper is all ready for publication, and I recommend acceptance after the authors address the points described below.

General comments:

First, I should say that I am not an experienced user of MELTS or the Magma Chamber Simulator. I therefore cannot quantitatively evaluate the results described in supplementary Table 3a or 3b, and can only state that the petrologic modeling and its results seem internally consistent and accurate based on what I know from reading previous work.

1. The paper would be improved somewhat if the very end provided a bit more

discussion about what the new results imply (if anything) about the extent / occurrence of EM2-derived compositions as a mantle isotopic component globally. Although extensive discussion is not appropriate for this paper's format, the end of the paper left me wondering what the authors were thinking are the new questions that have emerged (Line 302 simply states - "the presence of trachytic melts representing the EM2 end-member composition raises important questions about the origin of melts sampling this source"). Recognizing that trachytes are not commonly thought of as directly derived mantle melts, and following on from the last parts of the discussion in the supplementary material, it seems to me that more should be said here. The authors also seem to (perhaps) imply the new results hint somehow at how long recycled sediment might reside in the mantle source of OIBs that have an EM2 signature in this end paragraph, but I am unclear how that would be any different from what has already been discussed in the literature for a few decades now along the lines of mantle component origin/residence. The connection of EM domains to the LLSVPs seems to try to bring this into the modern picture of mantle dynamics (line 307). But it seems this last paragraph has mixed a few ideas together, and maybe missed an opportunity for discussing how the new results and modeling should influence our thinking about the regional or global origin and extent of the EM2 composition.

Response: To address this comment and partly the comment below, we have added more interpretation to the main text in the last 4 paragraphs. We now address in the main text much of what was presented in the supplementary text, dealing with the different processes that can create silicic, trachytic melts. We have removed the second and third model scenarios from the Supplementary text because it is now redundant. As stated in the updated letter to the editor, there is a companion paper to this manuscript that is in revision at G-cubed in which we go into detail about the processes that could have created this trachytic melt. Therefore, we deem it redundant to repeat this discussion and instead refer the reviewers to this paper (attached with this manuscript). Why this extreme EM2 signature remains so localized to Samoa remains a question that cannot be answered by this study (and we now state in the last paragraph). Lastly, we do not mean to imply in the last few sentences that our new results can say anything about residence times of recycled sediment in the mantle sources of OIBs. We simply aim to connect the fact that results of our study, consistent with previous studies, show that large fractions of recycled sediment must be present in the source of EM2 lavas in order to explain our mixing calculations. This suggests that since EM mantle domains are correlated geographically with LLSVPs, that sediment-rich material can reside in LLSVPs on the order of geologic timescales to be sampled in large quantities by the Samoan plume.

2. Following on from comment 1, it seems appropriate to point out somewhere in the paper the occurrence of somewhat unusual, high silica melts at other ocean islands that in some ways resemble the trachytic end-member melts proposed in this study. Examples include Réunion (Ludden (1976) Canadian Mineralogist 16, pp. 265-276), La Palma, Canary Islands (Wulff-Peterson et al, (1999) Contrib. Mineral. Petrol. 137, pp. 59-72), Kerguelen (Schiano et al. 1994, EPSL 123, 167-178), and Grande Comore (Schiano et al. 1998, EPSL 160, 537-550). There are probably other cases of which I am unaware. One of the above occurrences (Réunion) may have originated by extreme differentiation within the crust; however, in most cases above the compositions are found as melt inclusions in mafic igneous phases, and have been suggested to ultimately originate by various processes that may include the formation of immiscible alkali-rich liquids in basaltic melts (Réunion), by reaction between host peridotite and infiltrating basaltic melts (La Palma), by metasomatism from exotic (non-

peridotitic) melts (Kerguelen), or as the very earliest melts derived from spinel peridotite (Grande Comore). I am not sure how much more can be made of this that stems from the present study (some ideas are broadly mentioned with regard to Samoa in the supplementary discussion of model scenarios 2 and 3, but other island occurrences are not really mentioned there). It seems the presence of these types of melts may be more widespread than recognized and should be mentioned.

Response: Thank you for the detailed feedback. We now detail the potential hypotheses for the formation of trachytic melts in the mantle in addition to suggesting future avenues of work that could help address some of these hypotheses directly (see lines 481-539). We cite the papers suggested by the reviewer in addition to other new citations. Again, the above types of scenarios are discussed in more detail in the G-cubed companion paper discussed in the previous response.

3. There are a number of terms used in the paper that are common to igneous geochemists, that made it a bit of scramble for me to look around in the paper (sometimes ahead) to try to really understand what precisely the authors meant. The terms include “evolved”, “mixing”, “hybridization” and “co-mingling” (these are most obvious in the discussion that begins on line 282). Is co-mingling just inefficient mixing, or no mixing at all, just their juxtaposition? How does hybridization fit in the order of these things? An extensive explanation is not needed, but I do think it would help if the authors stated more precisely what they envision these terms mean when they first use some of them.

Response: Definitions of these terms have been added to this section for clarity. The discussion at lines 441-447 says the following: “It is expected that mixing of two (or more) magmas would produce crystal populations with heterogeneous major, trace, and $^{87}\text{Sr}/^{86}\text{Sr}$ and $^{143}\text{Nd}/^{144}\text{Nd}$ compositions since the hybridization (complete mixing or homogenization of two or more magmas) of the mixing endmember magmas is not instantaneous and most likely incomplete, even after decades of magma co-mingling¹³ (where co-mingling refers to incomplete mixing between two or more magmas so as there are discrete compositional bands in the actively mixing magmas).”

Minor comments:

1. Abstract. It is stated in the paper that “a direct comparison of isotopic heterogeneity observed in clinopyroxene and plagioclase within a single hand specimen provides new constraints on the source of isotopic heterogeneity...” (line 96-97; with further important details described on lines 163-179)) but this cpx-plag comparison is not mentioned in the abstract, and instead it only mentions clinopyroxene. Line 50 would have been more clear (to me at least) if the terms evolved = “evolved (trachytic)” or simply “silicic” and less evolved = “less evolved (basanitic)” or simply “mafic”. Also, the last sentence of the abstract would be better used to describe what the important new constraint(s) is(are) based on the new observations rather than the general statement that is currently provided.

Response: To address the first point, a few words have been added to the abstract to point out that a comparison with the previously published plagioclase isotope data is used in conjunction with the new cpx analyses to help place better constraints on the origin of EM2. This new wording was added at lines 72-76 and now reads: “In addition to previously published isotopic variability in plagioclase grains from the same sample, the new results provide insight into the composition of melts derived from an EM2

source, help provide constraints on the composition of the EM2 mantle domain, and document the fate of sediment recycled into the Earth's mantle.”

To address the second point, for clarification, the words “silicic” and “mafic” have been added to the abstract after the words evolved and less evolved, respectively at lines 52-53.

To address the last point, the last sentence of the abstract has been changed to provide more insight into the actual meaning of the results of this study. The last sentence of the abstract (lines 72-76) now reads: “In addition to previously published isotopic variability in plagioclase grains from the same sample, the new results provide insight into the composition of melts derived from an EM2 source, help provide constraints on the composition of the EM2 mantle domain, and document the fate of sediment recycled into the Earth's mantle.”

2. Line 147 (caption to figure 1) should add that the 9 of 17 crystals analyzed for Nd isotopes encompass the full range of $^{87}\text{Sr}/^{86}\text{Sr}$.

Response: The following words have been added to the Figure 1 caption at lines 202-203:

“However, the 9 clinopyroxenes analyzed for both Sr and Nd isotopes encompass the full range of $^{87}\text{Sr}/^{86}\text{Sr}$ observed in this study”

3. Line 260. I was confused by the statement of how “no coherent zoning pattern” in clinopyroxene is evidence for at least one magma mixing event. Based on Fig. S6, I think it would be better described in this sentence (L260) as an “irregular zoning pattern” (at least that seems less confusing to me).

Response: Thank you for the suggestion, we have changed “no coherent zoning pattern” to “an irregular zoning pattern” at line 405.

4. I found the discussion from line 431 to 443 rather essential to my acceptance that the modeling seemed internally consistent with a reading of the rest of the paper. In particular, the observation that the MgO in each end-member is confined to a rather narrow range of values based on the lava compositions themselves, and the linearity of the SiO₂-MgO mixing that extends between the other island data and approaches a fictive end-member with 0% MgO. I think it would be useful to include mention of these points in the main text where the modeling is outlined rather than just the supplementary material.

Response: We do briefly mention these points in the main text of the manuscript, however for further clarification, we have added the following sentences to the main text at lines 339-344: “In comparing SiO₂ versus MgO concentrations, the ALIA-D115 lavas show a linear mixing array that extends from a group of mafic Samoan shield lavas to a hypothetical endmember extending towards 0 wt% MgO. Thus, an averaged Samoan shield lava value is used for the MgO concentration of the mafic mixing endmember and we test 3 different MgO concentrations for the silicic mixing endmember from the lowest value observed in ALIA-D115 lavas (3.8 wt% MgO) to 0.5 wt% MgO (see Methods for more details).”

The use of K instead of Mg makes sense when looking at some of the other elements because it should help to reduce spurious correlation effects. However, whenever

elemental ratios are plotted using a common denominator, the possibility of seeing a spurious correlation cannot always be eliminated unless other criteria are met (e.g. Jackson & Somers 1991, The spectre of spurious correlations, *Oecologia* 86, 147-151). Some of the trends seen in Fig. S4 are not as highly correlated as others (e.g., involving K₂O, MnO, FeO and P₂O₅) and some spurious effects may contribute to some of them. This does not call into question the suitability of the approach, which clearly stems from the observations of whole rock isotopes and MgO and SiO₂ as mentioned. I do think the authors might want to critically assess how much of the simple binary mixing vs. (seemingly minor) spurious effects are responsible for the different elements they are showing in this figure. Along the same line, figure S5 shows the companion binary mixing plots (involving the elemental ratios) to the ⁸⁷Sr/⁸⁶Sr-¹⁴³Nd/¹⁴⁴Nd diagram. Simple binary mixing should also be (even more) evident on plots including the reciprocal elemental ratios (1/Nd and 1/Sr), but these are not discussed in the supplement. If simple binary mixing is responsible, then these plots also allow an extreme limit to be placed on the isotope composition of the end-member where the reciprocal concentration (i.e., 1/Nd or 1/Sr) goes to zero. Obviously, this will be complicated by the fractional crystallization of plagioclase as discussed in the main text, but that effect also influences the other supplementary figures mentioned above that involve Sr.

Response: It is true that some correlations are not as highly correlated as others, this is likely related to fractionation effects, however, it doesn't really matter what elements you choose as the denominator in these calculations (we have chosen to use MgO and K₂O), the derived endmember compositions will be quite similar. The values change very slightly based on how good the correlations are, but not significantly. As an example, we used the ratio of Sr/K₂O vs. 1/K₂O to derive Sr concentrations in the two endmembers with a correlation coefficient of 0.96; the resultant Sr concentration in the mafic endmember was 712 ppm and 380 ppm in the silicic endmember. If we had instead performed the same calculation with Sr/MgO versus 1/MgO, the correlation coefficient is 0.85 and the Sr concentration in the mafic endmember would be 726 ppm and the silicic endmember would be 360 ppm. These differences are comparable to analytical uncertainties for real measurements. Thus, spurious correlations are not affecting our results in a significant way. Further, with regard to why we did not plot Sr and Nd isotopes versus 1/Sr and 1/Nd, respectively, this is because it is not technically the proper companion plot, based on mixing theory (see Langmuir et al., 1976). Indeed, the correlation still holds true when plotting ⁸⁷Sr/⁸⁶Sr or ¹⁴³Nd/¹⁴⁴Nd versus 1/Sr or 1/Nd, respectively (we have done this). However, in our case this correlation is actually slightly worse than with the correct companion plot so we found no reason to also include these plots in the paper.

5. Line 841 compares equilibrium vs. batch melting as though they are different. Maybe fractional melting (or critical melting) vs. batch melting was intended?

Response: We did intend equilibrium vs. batch melting where *equilibrium melting* is melting in which the partial melt and residual solid remain in equilibrium at all times during progressive partial fusion. That is, there is no separation of melt (at any stage) from residual crystals. In contrast, during *fractional melting* each *infinitesimal* increment of melt is removed upon generation by progressive partial fusion. In batch melting (perhaps we should call this *incremental batch melting*), melt remains in equilibrium with its crystal residue *until some level of melt accumulation is attained* (e.g., 10% incremental batch partial melting) after which melt is removed and the (now) solid residue can undergo further partial melting.

To define incremental batch melting (which we simply called batch melting but have modified now in the text to incremental batch melting), the fraction of melt that triggers removal is specified. For example, if we specified *10% incremental batch melting*, one starts at the solidus and melt is generated until the mass fraction of melt is 0.1. At that instant, all the melt is removed. If partial melting continued and it was again *10 % incremental batch melting*, then this new “stage two” (initially all-solid) source would be partially melted to the extent of 10% and *that* melt would be removed.

In incremental batch melting, arbitrary (but specified explicitly) fractions of melt remain in equilibrium with the residual solid until the specified ‘critical’ fraction is met. The important difference between incremental batch melting and equilibrium melting is that in incremental batch melting, the bulk composition of the system is changing during the incremental steps whereas in *equilibrium melting*, the bulk composition of the system is fixed no matter what the extent of melting may be.

We have added the adjective **incremental** and a sentence defining our usage so there should be no confusion.

The following sentence has been added for clarification where we mention equilibrium vs. batch melting (lines 1355-1359): “...type of melting (i.e., equilibrium versus incremental batch; where equilibrium melting involves partial melting in which the partial melt and solid are in complete equilibrium at all times and incremental batch melting involves achieving a certain melt fraction in stages, where specified fractions of melt are incrementally removed from the residual solid until the total desired melt fraction is met)”

5. The acknowledgments include E.A.P. (not one of the authors ?).

Response: Thank you for catching this. This should read A.A.P and has been changed.

David Graham
April 21, 2020

Reviewer #3 (Remarks to the Author):

This manuscript by Adams et al., reports impressive coupled Sr-Nd isotopic data on a set of clinopyroxene phenocrysts from a dredge sample from Samoa. The major claim is that the isotopic variability results from binary mixing of a mafic and an evolved trachytic melt, the latter which is suggested to represent a EM2 component that includes old recycled sediments. The suggestion that the endmember melt has a trachytic compositions is novel.

The high-quality data are exceptional and intriguing, and their implications are of significant interest to the community. This to me justifies publishing this paper in Nature Communications. The manuscript is well-written and the model calculations are described in enough detail as to understand and reproduce them. The way the data is interpreted and discussed currently, however, can in my opinion be more robust and could do with some clarification and restructuring. I have a few more major points that I raise here, more detailed comments and minor edits can be found in the annotated manuscript.

1) The authors present coupled major and trace element and Sr-Nd isotopic data of clinopyroxenes. However, in their interpretation and mixing models they use the bulk lava compositions rather than the clinopyroxene data that is argued to record the mixing process in more detail. Can they not use the cpx major and trace element data to calculate theoretical parental melts? I can guess what could be the reason that would not work very well, but think it would be fair to address this in the paper. I assume the relations between the elemental data and the isotopes in the clinopyroxenes are less well defined or absent? Why? Can part of this be explained by the fact that the LA-ICPMS data is not representative for the bulk cpx and would it maybe have been better to do ICPMS on a aliquot of the dissolved sample?

Response: This is a really great point. We did actually grapple with this and the reason we did not use the clinopyroxenes is for the exact reason you have pointed out. Relationships between major and trace elements versus the isotopes were not very well defined and thus, we could not use them to reconstruct the endmembers. This is almost certainly due to the fact that the major and trace element measurements in the clinopyroxenes were performed by spot analyses, whereas the isotopes were measured by dissolution. These clinopyroxenes are quite zoned in terms of major and trace elements, and while both the major and trace element data are averages of multiple spot analyses, we still obtain ill-defined mixing trends for the clinopyroxene data with the exception of Sr and Nd isotope space. Further, the mixing trends with clinopyroxene can be further obfuscated by the fact that they are affected by differential partitioning of certain elements in the crystal versus the liquid, whereas the whole rock elemental compositions are less affected by this process (and isotopes of whole rock and clinopyroxenes are unaffected by this process). However, we agree that addressing this fact in the paper is a good idea. The following sentences have been added at lines 276-284 “Note that we do not use the clinopyroxenes for reconstructing the endmembers because they show ill-defined mixing trends with respect to major and trace elements. This is likely due to 1) fractionation effects during their growth and 2) the clinopyroxenes being zoned such that averaged spot analyses of trace element concentrations (determined by averaging multiple LA-ICP-MS spots on each crystal; see Methods) will not correspond directly with grain surface major elements (obtained by extracting average major element concentrations from EPMA maps; see Methods) and will thus produce spurious trends, especially when compared to radiogenic isotopes (which were measured on dissolved crystals). However, these issues will not affect the isotopes, which is why the mixing trend is evident for $^{87}\text{Sr}/^{86}\text{Sr}$ and $^{143}\text{Nd}/^{144}\text{Nd}$ ”

2) The mixing model is built on relations between Sr-Nd isotope compositions and major element compositions, with MgO being an important fixing point. Major elements are known to be prone to crystallisation of major phases such as olivine, pyroxene or feldspar, so to me it would have been more robust to use trace element ratios to avoid the influence of fractionation. The model seems to work for Sr/Nd ratios, but I am curious to see if other ratios of highly incompatible elements that are commonly used as source indicators also work for the model; e.g. Ce/Pb, Nb/U, Nb/La, Th/Nb etc

Response: Per request of this reviewer, we have demonstrated that other elements, when normalized to MgO concentrations, can be derived by this method as well. In particular, we focused on new calculations of Nb and Th. We disregard U because U is compromised in the submarine ALIA lavas due to weathering, so we exclude the Nb/U ratio. Pb is fluid mobile and also often compromised by alteration in submarine environments, so we do not use the Ce/Pb ratio. Further, we exclude the Nb/La ratio because Nb/La is influenced by melting processes more than Nb/Th (see Hofmann, 2014 Figure 18). Nb/Th can reliably track mantle source characteristics so we show a plot of Nb/Th vs. $^{87}\text{Sr}/^{86}\text{Sr}$ in Supplementary

Figure 8 and describe this plot briefly in the main text. The following sentence has been added to the main text to address these additional calculations (lines 311-314): “In addition, as proof of concept, this linear regression method was applied to two other select trace elements (Nb, Th) pertinent to generating a ratio useful for tracking source mantle characteristics (Nb/Th; Hofmann, 2014; see Supplementary Figure 8 and Supplementary Table 2 for concentrations in the endmembers).” Supplementary Figure 8 is described briefly at lines 408-414 that state: “Supplementary Figure 8 shows that Nb/Th of the ALIA-D115 lavas decreases with increasing $^{87}\text{Sr}/^{86}\text{Sr}$, consistent with magma mixing between a mafic, low $^{87}\text{Sr}/^{86}\text{Sr}$ (high Nb/Th) source and one that has high $^{87}\text{Sr}/^{86}\text{Sr}$ and low Nb/Th ratios (Nb/Th=1.8 for the low MgO EM2-derived endmember of this study) similar to upper continental crust (Nb/Th=1.1³⁵) or average sediments (Nb/Th=1.2³⁶). However, as stated above, neither of which are modern materials, but rather ancient subducted materials”

3) Even though I agree that the data seems to suggest some sort of binary mixing process; I think the authors should address in more detail what the physical implications of their theoretical model are. For example the conditions of magma genesis and mixing are not addressed in detail in the text. First of all, some sentence(s) should be included to argue why it cannot be AFC in the crust (absence of CC and depth constraints?).

Response: We are not sure exactly what the reviewer is requesting, but we make a good-faith effort to address the comment based on our understanding of the comment, which seems to focus on addressing assimilation of continental crust at shallow levels. It has been shown at Samoa through decades of work that the high $^{87}\text{Sr}/^{86}\text{Sr}$ mixing endmember cannot be derived by shallow crustal contamination. This is because the most radiogenic lava from Samoa, and the Samoa clinopyroxenes in this study, are much more radiogenic than is generally achieved by oceanic crust, which rarely reaches $^{87}\text{Sr}/^{86}\text{Sr}$ above that of seawater. Further, it has been shown in previous studies that the high $^{87}\text{Sr}/^{86}\text{Sr}$ cannot be related to assimilation of modern sediments either because the Pb isotopic compositions of modern global sediments, as well as sediments in the Samoan region, show a non-overlapping trend with increasing $^{87}\text{Sr}/^{86}\text{Sr}$ that diverges away from the Pb isotopic compositions observed in the ALIA lavas, which are the lava discussed in this study (see Jackson et al., Nature, 2007). To address this important point, a few sentences have been added to lines 401-408 summarizing the above statements.

But also, the claim for the geochemical relations to reflect binary mixing of mantle derived melts requires how that could physically work. What are the implications for the mechanism of melt generation, mixing and extraction? How can we have had a system where 80% of trachyte melt mixed with 20% mafic melt to generate the most extreme cpx? Did the systems stay isolated until cpx crystallised? What does it infer with respect to the size and distribution of recycled components? Why do we only see the heterogeneity for this locality? Some of that discussion is now in the supplementary text, but should be expanded by using related literature (currently there is quite a large amount of self-citation) and moved to the body text. I think there is no strict word limit that would not allow that.

Response: Thank you for the feedback. The supplementary text is essentially dedicated to discussing these questions. However, because there is a slightly longer format for Nature Communications, and per the request of the other reviewer also, we have added more substantial interpretation to the main text in order to address as many of these points as we could. Additionally, as stated in a previous response, a substantial amount of interpretation with regard to the physicality of transport of trachytic melts in the mantle in addition to the mixing process is addressed in the companion paper in revision at G-cubed and has been attached with the resubmission of this manuscript. However, a fair amount of discussion has

been added to the last few paragraphs of this manuscript to address part of this reviewer's comment more fully. See lines 398-484 for the complete discussion. We discuss a potential magma mixing scenario that could explain the cpx variability with regard to $^{87}\text{Sr}/^{86}\text{Sr}$ in addition to scenarios that could create trachytic melt mixing endmember.

4) There is some important petrographical information that is either presented very late in the manuscript (trachytic melt inclusions in the cpx) or in the supplement alone (plag mineral inclusions within the cpx). Some of that data is used as an argument to support the interpretation (line 297), but I guess the petrography should rather be the fundament for the interpretations.

Response: We are not sure we understand this comment. As the reviewer notes, we include petrographical information in the text, and we agree that we use some of this petrographical data to support our interpretations. Therefore, it is not clear what the reviewer is looking for.

I hope my comments and suggestions will help to improve a revised version of the manuscript to be accepted in this journal.

Janne Koornneef

REVIEWERS' COMMENTS

Reviewer #1 (Remarks to the Author):

The authors have addressed my earlier comments in their response to my review. I found the revised manuscript to be coherent and well argued in its presentation, and the broader implications of the presence at ocean island hotspots, of highly evolved and radiogenic mantle-derived melts, are now more clearly outlined. This study will stimulate new research into the origin of such melts, both theoretically and experimentally. I believe the paper should be published in a timely manner.

David Graham

Reviewer #3 (Remarks to the Author):

I have read the revised paper and it has much improved with the implementations made. I am happy with the way the main points of my review were addressed, despite that many of the minor -less important- comments in the annotated PDF have not been replied to or addressed. If the editor complies, I agree to accept the manuscript after these final cosmetic changes are made.

- Add a sentence with the larger implications to the end of the introduction
- Fig 1: Symbols for cpx and bulk rocks are both squares; change symbol of either one to a different type.
- Line 134: more significant curvature than
- Line 367: give rise to silicic with: add melt

Janne Koornneef

Dear Editor,

Below are responses to these final reviewer comments. Changes made to the manuscript due to these reviewer comments and the editorial requests have been highlighted in yellow within the revised manuscript.

Thank you!

REVIEWERS' COMMENTS

Reviewer #1 (Remarks to the Author):

The authors have addressed my earlier comments in their response to my review. I found the revised manuscript to be coherent and well argued in its presentation, and the broader implications of the presence at ocean island hotspots, of highly evolved and radiogenic mantle-derived melts, are now more clearly outlined. This study will stimulate new research into the origin of such melts, both theoretically and experimentally. I believe the paper should be published in a timely manner.

David Graham

Response: Thank you for the helpful comments!

Reviewer #3 (Remarks to the Author):

I have read the revised paper and it has much improved with the implementations made. I am happy with the way the main points of my review were addressed, despite that many of the minor -less important- comments in the annotated PDF have not been replied to or addressed. If the editor complies, I agree to accept the manuscript after these final cosmetic changes are made.

- Add a sentence with the larger implications to the end of the introduction

Response: A sentence has been added at line 65-67 stating the following: "Moreover, these new isotopic constraints help shed light on the origin of terrigenous sediment in the Samoan plume source and, thus, the deep-mantle residence of subducted terrigenous material in large low shear wave velocity provinces (LLSVPs)."

- Fig 1: Symbols for cpx and bulk rocks are both squares; change symbol of either one to a different type.

Response: Fixed. We changed the squares for the cpx to squares with crosses in them.

- Line 134: more significant curvature than

Response: Fixed. See line 90.

- Line 367: give rise to silicic with: add melt

Response: Thank you for catching this. This part of the sentence now reads: “partial melting of this source could give rise to silicic melts with radiogenic $^{87}\text{Sr}/^{86}\text{Sr}$.” See line 262

Janne Koornneef